# Degradability of Polylactide in Natural Aqueous Environments

**Katarzyna Krasowska** * **and Aleksandra Heimowska** 

Department of Industrial Products Quality and Chemistry, Faculty of Management and Quality Science, Gdynia Maritime University, 81-87 Morska Str., 81-225 Gdynia, Poland
* Correspondence: k.krasowska@wznj.umg.edu.pl

**Abstract:** This study aims to estimate the degradation process of polylactide (PLA) in natural aqueous environments. The biological degradation of PLA took place in the Baltic Sea and in the natural pond over a period of 1 to 16 months. The characteristic abiotic parameters of both environments were monitored during incubation time, and their influence on the PLA degradation was discussed. The changes in weight, chemical structure, mechanical properties and surface morphology of investigated samples were also tested during incubation. The obtained results indicate that polylactide is not very susceptible to an enzymatic attack of microorganisms present in natural aqueous environments.

**Keywords:** polylactide; degradation; natural environment; sea water; pond

## 1. Introduction

Polymer materials are more commonly used in various everyday applications, especially in packaging. High plastic consumption generates plastic waste responsible for environmental pollution. Plastic waste that is not recycled, incinerated or landfilled is then found in nature and pollutes seas, rivers, oceans, lakes, beaches, forests, etc. Nowadays, important challenges for the future are sustainable consumption and use of plastics, better management of polymer lifetimes, and limiting the use of traditional polymers. The problem of environmental plastic pollution has been addressed by several authors, but thus, far, no one has discovered a complete solution.

Biodegradable polymers are seen as a promising potential alternative to improve the global plastic waste problem in the natural environment. In contrast to traditional polymers, biodegradable polymers are susceptible to biological degradation once they end up in the natural environment.

In general, the process of polymer biological degradation can be divided into the following steps: biodeterioration, depolymerization, bioassimilation, and mineralization [1,2]. Polymer biodegradation is the consequence of a number of physical, chemical and biological processes related to the destruction of polymers [3]. In the first step, the formation of biofilm on the polymer surface leads to superficial degradation. The large particles of polymeric material are fragmented into smaller ones (e.g., into the microplastic form) [4,5]. The microorganisms of the biofilm excrete extracellular enzymes, which catalyze the depolymerization of the polymer chain into smaller molecules (monomers, dimers and oligomers) before they can be assimilated through the cellular membrane. In the final step of polymer biological degradation, metabolites are mineralized, and end-products like carbon dioxide, water and biomass (or methane under anaerobic conditions) are formed and released into the environment.

This multistage biological degradation process is influenced by the synergistic actions of the environment's physical, chemical and biological factors [6].

Additionally, biodegradable polymers have been found to degrade more rapidly if a combination of microbes is used rather than one specific microbe. Thus, the presence of large varieties of microbes in the natural environment supports faster biodegradation [7,8].

In the family of biodegradable polymers, polylactide (PLA) is one of the most frequently used polyesters. This is mainly due to its many favorable properties, including its easy availability, relatively good strength, biocompatibility, and biodegradability. Besides starch blends, which are the most popular and account for 35% of the global production capacities for biodegradable plastics, PLA is also produced commercially. Production capacities of PLA are also predicted to grow by 83% by 2027 compared to 2022 [9]. The PLA is now at a price level similar to that of polystyrene [10].

Most of the articles available in the literature describe the degradation of PLA in controlled enzymatic tests or laboratory tests in simulated environments [11–18]. However, there is an absence of standards and test methods for evaluating the biodegradability of plastic materials within freshwater ecosystems (including lakes, ponds, streams and rivers) and most marine environments [2,19].

The test temperatures used in the laboratory are frequently higher than those in natural aquatic environments despite the ability of temperature to strongly influence the taxonomic composition and metabolic activities of microbial communities. Furthermore, the maximal test durations of published laboratory tests are likely to be too low for assessing the breakdown of certain polymer types due to rates of biodegradation being comparatively low within these environments [20,21].

The laboratory tests allow for the understanding of the working principles of enzymes and the general mechanism of polymer biodegradation processes. Yet, as proof of biodegradability alone, they are not enough and should always be conducted together with polymer degradation in a natural environment under uncontrolled conditions.

The natural environments are more complicated, very often require a longer time of polymer incubation and do not guarantee visible effects of decomposition. Therefore, environmental polymer degradation can go on for several days or months and even several years.

The negative effects of polymers on the natural environment and human health and the problems associated with their resistance to degradation have significantly increased over the past decades. Currently, special attention is given to very small pieces of polymer particles (smaller than 5 mm in their longest dimension fall), which are referred to as microplastics. Hazardous substances and microplastics are probably the predominant anthropogenic sources in riverine load and land runoff into natural aqueous environments [22]. Microplastics are commonly divided into primary and secondary. These secondary microplastics result from the fragmentation of larger plastic debris, either during use or during environmental destruction. Fragmentation can occur through photodegradation by UV light, hydrolysis in the presence of water, (thermal) oxidation, physical abrasion in sediments and soils but also by waves, and biological degradation by organisms [23,24].

Although there are a significant number of studies concerning the biological degradation of PLA in industrial compost, activated sludge, home compost or cultivating soil, far fewer studies focus on PLA degradation in aqueous environments (especially in seawater and freshwater) [25–33]. Therefore, a significant lack of knowledge of PLA degradability exists in aqueous natural environments such as lake, river, pond and marine environment. Hence, it is of interest to study the changes in PLA properties in "real" environments to assess whether they can cause the creation of secondary microplastics in these environments. The main intention of this paper is estimating the susceptibility to degradation of PLA during incubation in the pond and in the Baltic Sea.

## 2. Materials and Methods

### 2.1. Environments

The environmental degradation of polymer materials took place in two natural aqueous environments: in the pond and in the Baltic Sea. The incubation of polymer samples in freshwater occurred in Rumia's pond and in seawater in Gdynia Harbour on the Norwegian Pier near the ship of the Maritime Search and Rescue Service.

The polymer samples were located in special perforated baskets, suspended from a rope at a depth of 2 m below the water surface of the Baltic Sea and Rumia's pond. The perforated basket structure allowed for the free movement of water and the access of microorganisms and enzymes dissolved in the aqueous environments to degrade the materials [34].

The temperature, pH, oxygen, and salt content of natural environments were monitored during the degradation process of investigated polymers.

For comparison, the degradation of polymer samples also took place in fresh water with sodium azide ($NaN_3$) and seawater with $NaN_3$ (0.195 $g/dm^3$) in a laboratory [35]. The sodium azide was added to water from the Baltic Sea and Rumia's pond to exclude the activity of microorganisms and evaluate the polymers' vulnerability to chemical hydrolysis. The PLA samples were located in two 60-L glass aquariums equipped with an aeration pump.

The incubation of investigated samples lasted for a period of up to 16 months in natural weather, depending on conditions.

### 2.2. Material

The commercial BIO PLA™ multilayer film was studied. The raw material was manufactured from NatureWorks®PLA (polylactic acid), a renewable polymer derived from cornstarch, an annually renewable product. BIO PLA™ is just one product from BioPak (Bondi Junction, Austalia). Poly(lactide) film (PLA), with the trade name "BIO 521," was degraded. This polymer consists of three layers as follows: sealable PLA, core PLA and sealable PLA [36].

The polymer film was cut into 150 × 20 mm rectangles. After incubation in water environments, the samples were washed, dried at room temperature and then were taken to investigations.

### 2.3. Methodology

The weight changes, macro- and microscopic observations, chemical structure and mechanical properties of PLA were investigated before and after degradation in the natural and aqueous laboratory environments. In the experiment, three to five samples were taken from the aquatic environment, and the arithmetic average of the measurements was used.

- **The macro and microscopic observations:** The polymer samples were observed on a macro- (naked eye) and micro-scale (microscopic observation). Macroscopic observations of the polymer surface were analyzed organoleptically using a FujiFilm S2500 HD camera. In contrast, the microscopic observations of the polymer structure were analyzed with the metallographic microscope, Nikon ALPHAPHOT-2YS2-H (Polish Optical Companies, Warsaw, Poland), linked to the photo camera Delta Optical DLT-Cam PRO 6.3MP USB 3.0. (Delta Optical, Gdansk, Poland). The micrographs were collected under transmitted light. The images of the polymer samples before and after degradation were compared.
- **The changes in weight**: The dried polymer samples were weighed on an electronic analytical balance RADWAG AS 160.X2 (Radwag, Radom, Poland) with a repeatability 0.1 mg. The weight of clean and dried polymer samples after incubation in the natural aqueous environments was compared with the one before incubation.
- **The changes in chemical structure:** Attenuated Total Reflectance Fourier Transform Infrared Spectroscopy (ATR-FTIR) was used to determine the characteristic groups of PLA. FTIR spectra were recorded with an attenuated total reflection (ATR Smart Orbit Accessory, Thermo Fisher Scientific, Madison, WI, USA) mode on a Nicolet 380 FTIR spectrometer (Thermo Scientific, Madison, WI, USA) with a diamond cell. A resolution of 4 $cm^{-1}$ and a scanning range from 600 to 4000 $cm^{-1}$ were applied, and 32 scans were taken for each measurement.
- **The changes in mechanical properties:** The maximum tensile strength (MPa) was measured at room temperature using The Tensile Testing Machine MultiTest 1-xt made

by Mecmesin (ITA Ltd. Poznan, Poland), according to PN-EN ISO 527-1, 3: 2018-19 Standard [37,38].

## 3. Results and Discussion

### 3.1. The Characteristic Parameters of the Natural Aqueous Environments

The abiotic and biotic parameters of the natural environment have an impact on the development of living organisms. The characteristic abiotic parameters of the aqueous environments were monitored during incubation time, and their influence on the degradation of PLA was discussed.

The characteristic parameters of both natural environments, pond and seawater, are shown in Table 1.

**Table 1.** The characteristic parameters of natural aqueous environments.

| Months | Pond Parameters [1] | | Baltic Sea Parameters [2] | | | |
|---|---|---|---|---|---|---|
| | Temperature [°C] | pH | Temperature [°C] | pH | Oxygen Content [cm³/dm³] | Salt Content [ppt] |
| July | 19 | 8.3 | 21 | 8.4 | 7.6 | 6.6 |
| August | 19 | 8.4 | 21 | 8.6 | 6.5 | 6.1 |
| September | 12 | 8.5 | 19 | 8.4 | 6.7 | 6.5 |
| October | 10 | 8.0 | 14 | 8.2 | 6.9 | 7.1 |
| January | 5 | 7.8 | 4 | 8.2 | 9.7 | 6.9 |
| May | 12 | 8.0 | 18 | 8.5 | 8.0 | 6.2 |
| July | - | - | 21 | 8.6 | 7.7 | 6.8 |
| November | - | - | 10 | 8.1 | 7.8 | 6.7 |

Notes: [1] Source: Own research [2] Source: Parameters received from the State Environmental Monitoring, Inspectorate of Environmental Protection.

The temperature of both natural environments was dependent on the weather conditions (winter and summer) and had been changing distinctively for the time of the experiment (from 4 °C to 21 °C). Looking at the parameters presented in Table 1, we can observe that the average temperature in the pond was approximately 13 °C and 16 °C in the Baltic Sea. The lower temperature was not preferable for enzymatic degradation (20–60 °C) in both natural aqueous environments [39].

If you look at Table 1, you will notice that the pH values were slightly alkaline and were above the upper limit of pH (5–8) recommended in the biodegradation process [38]. The average pH values were on the same level; the pond was 8.2 and 8.3 in the Baltic Sea. Only the strong sunlight during the summer months and the development of photosynthetic organisms are able to increase of pH of natural waters (Table 1) [40].

The water has the capacity to dissolve gases. This solubility decreases with increasing temperature [41]. During the winter months of incubation of PLA samples, we could observe the lowest temperature and the highest oxygen content (January 9.7 cm³/dm³). These conditions could influence the activity of oxidizing enzymes responsible for oxidation. It could be expected that these conditions had an influence on the development of aerobic epilithic bacteria. The metabolism of these microorganisms probably caused the decrease in oxygen content in the summer months (August 6.5 cm³/dm³).

The salinity of the Baltic Sea was in the range of 6.1–7.1 ppt (Table 1). The salt content of the Baltic Sea changes naturally and depends on the tributary of rivers or heavy rainfalls [42].

It is common knowledge that the pond and sea are very specific and complicated natural environments where microorganisms, animals, salt, sunlight, fluctuation of water, rain etc., play a vital part in the degradation process in nature [43]. The relatively low temperature and alkalinity of both natural environments (Table 1) could influence the activity of psychrotrophic bacteria, which can adapt to such changing conditions.

The incubation of PLA samples in the laboratory in two liquid mediums (pond water and sea water) with sodium azide ($NaN_3$) was performed at a stable temperature (about 21 °C) and under alkaline conditions (pH of approximately 8.0).

### 3.2. The Changes in Polylactide during Environmental Degradation

The environmental degradation of investigated PLA samples was assessed visually at first. Figure 1 presents macrographs of the polymer surfaces before and after degradation in the natural aqueous environments.

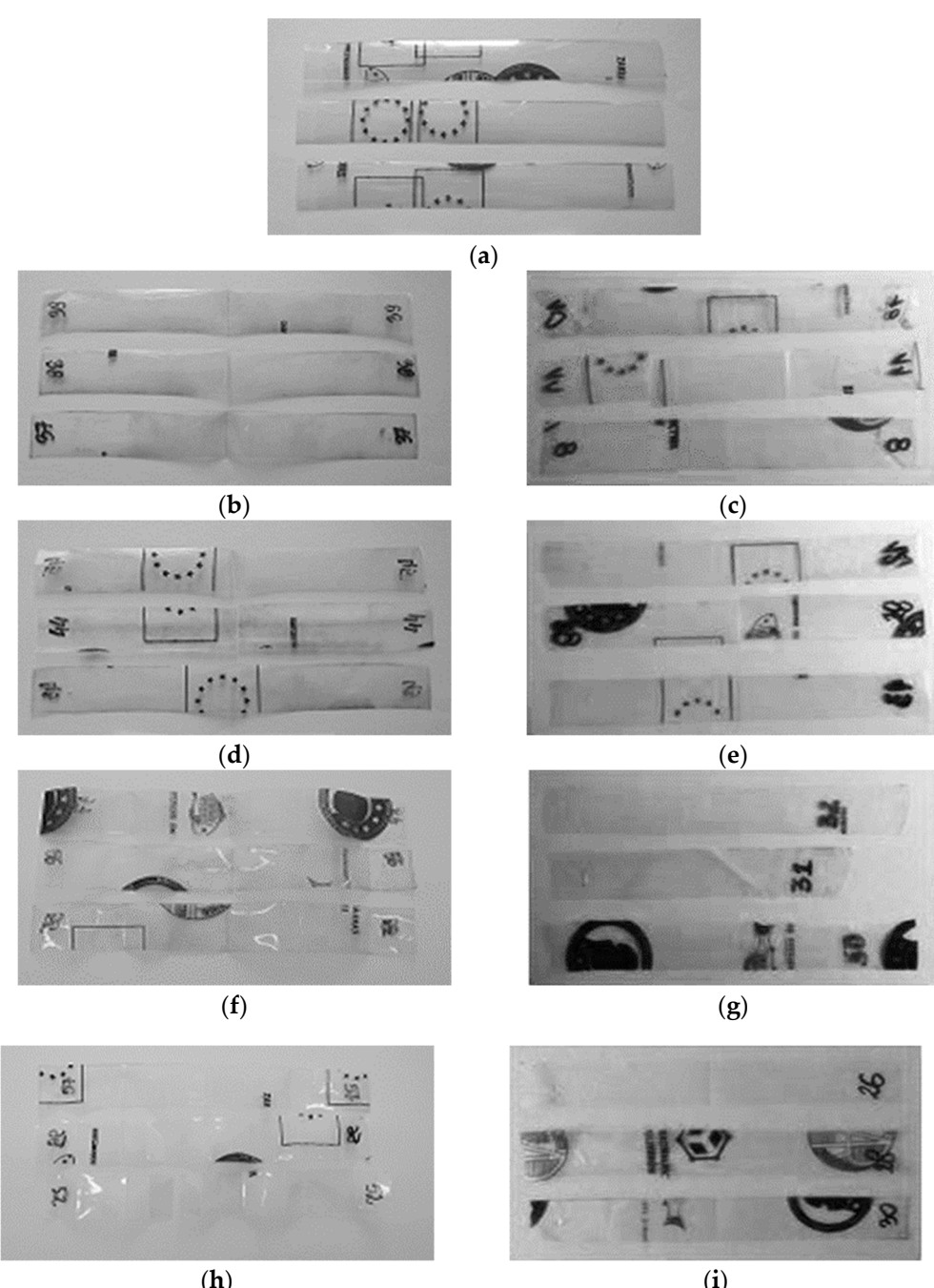

**Figure 1.** Macrographs of PLA samples before and after incubation in different aqueous environments: (**a**) before degradation; (**b**) 3 months in the pond; (**c**) 3 months in sea; (**d**) 6 months in the pond water; (**e**) 6 months in the seawater; (**f**) 12 months in the pond; (**g**) 16 months in the seawater; (**h**) 12 months in pond water + NaN$_3$; (**i**) 16 months in the seawater + NaN$_3$. Source: Own research.

The macroscopic changes on the PLA surface after incubation in both natural environments indicate their very slow environmental degradation (Figure 1). After incubation in the pond and sea, the slightly dark brown places on the PLA surface could be observed. It

was a consequence of small amounts of microorganism activity and, gradually, the formation of biofilm on the PLA surface. There were not any holes and fragmentation of PLA samples during all incubation time in natural aqueous environments.

No visible changes on the surface of PLA samples after incubation in both laboratory waters with sodium azide could be observed. Due to the absence of microorganisms in laboratory tests, only chemical hydrolysis could be expected.

The macroscopic observations were in good agreement with changes in the weight of polymer samples. The results of weight changes in the investigated PLA samples are presented in Figure 2.

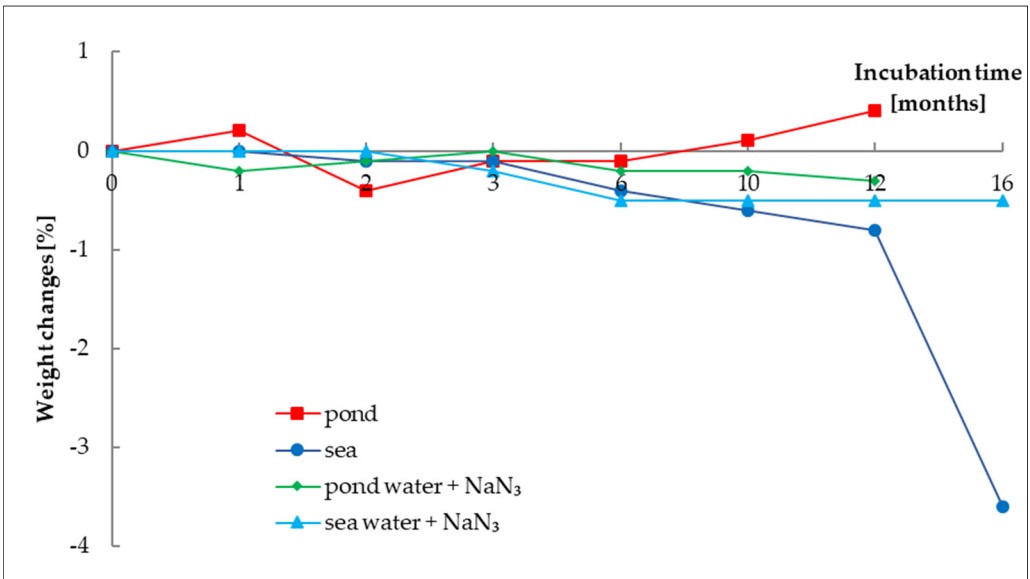

**Figure 2.** Weight changes [%] of PLA samples after incubation in different aqueous environments. Source: Own research.

Figure 2 compares the weight changes in PLA in a pond, sea and two laboratory tests. During all incubation times, the weight changes in PLA samples were not substantially detected in all of the aqueous environments during a 12-month incubation period. After 16 months of exposure to the Baltic Sea, a weight loss of approximately 3.5% was observed. The slight acceleration of PLA mass loss in seawater between the 12th and 16th months is most likely due to degradation in the summer months (May–November), where seawater temperatures were higher than in the winter months.

No significant weight losses of the PLA samples during incubation in both natural environments suggested that the pond and sea were not sufficiently microbial active for PLA degradation and confirmed sufficient resistance of PLA to degradation in aqueous environments [25–30]. Another reason for the lack of weight losses in PLA was the low average temperature of natural environments during the incubation time. It is known that temperature plays a key role in determining the rate of PLA degradation. The degradation rate is enhanced greatly at temperatures above or near the glass transition temperature of PLA (above 50 °C) [44,45].

Comparing the curves presented in Figure 2, we can observe insignificant weight changes in PLA during degradation in two laboratory tests. The minor changes may be explained by nonenzymatic hydrolytic ester cleavage during the incubation time. As a consequence of chemical hydrolysis, the changes in molecular weight should be noted and confirmed by measurements of the molecular weight of investigated PLA samples.

The degradation progress of PLA was also studied by assessing the change in its chemical structure after 12 months of incubation in different aqueous environments (Figures 3–5).

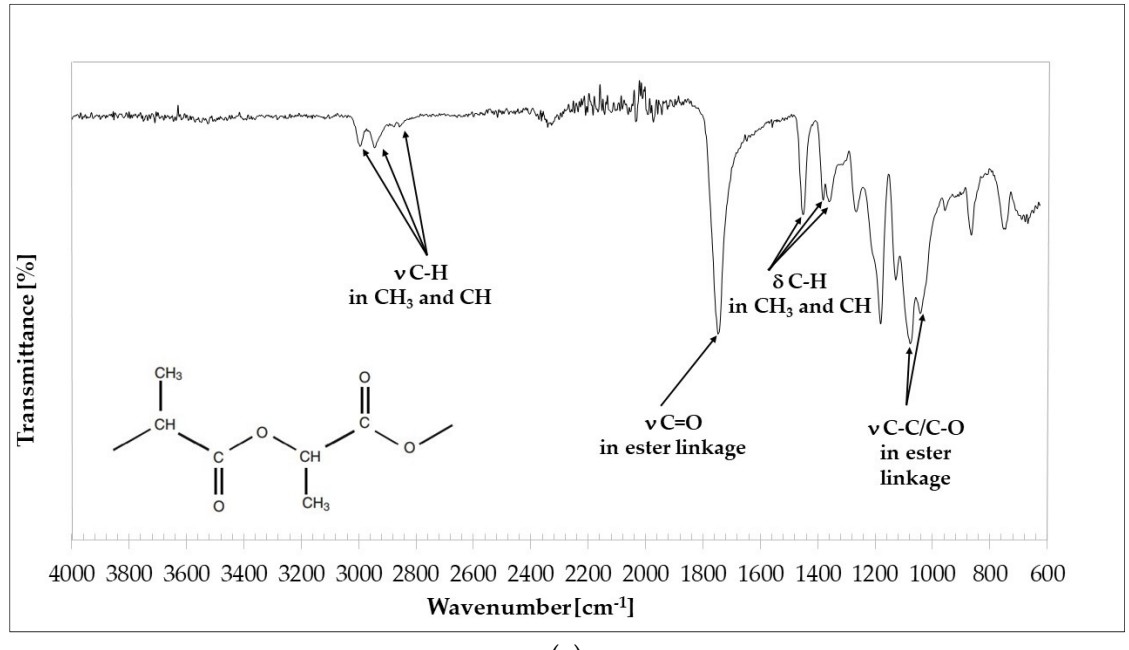

(**a**)

(**b**)

**Figure 3.** ATR-FTIR spectra of the PLA sample in the wavenumber of 600–4000 cm$^{-1}$: (**a**) before degradation; (**b**) after 12 months of degradation in aqueous environments. Source: Own research.

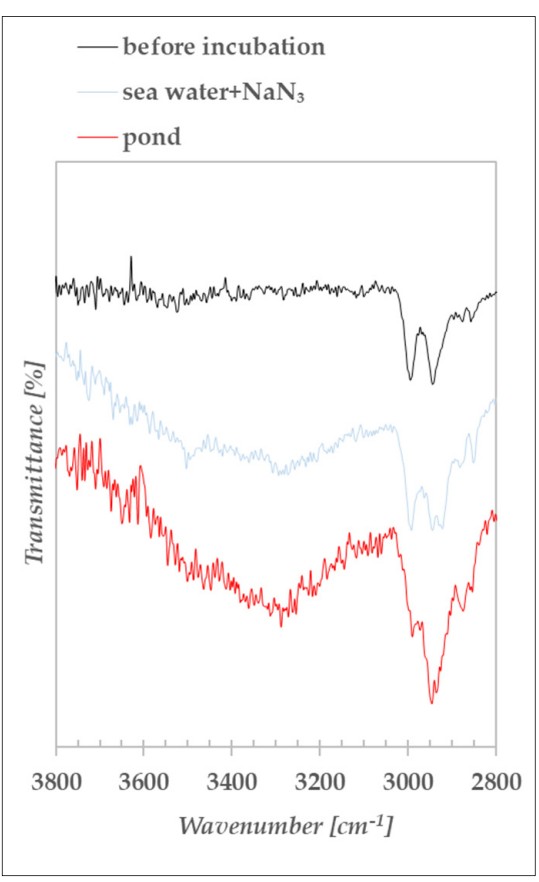

**Figure 4.** ATR-FTIR spectra of the PLA sample before degradation and after 12 months incubation in pond and seawater with $NaN_3$ as the wavenumber of 2800–3800 $cm^{-1}$. Source: Own research.

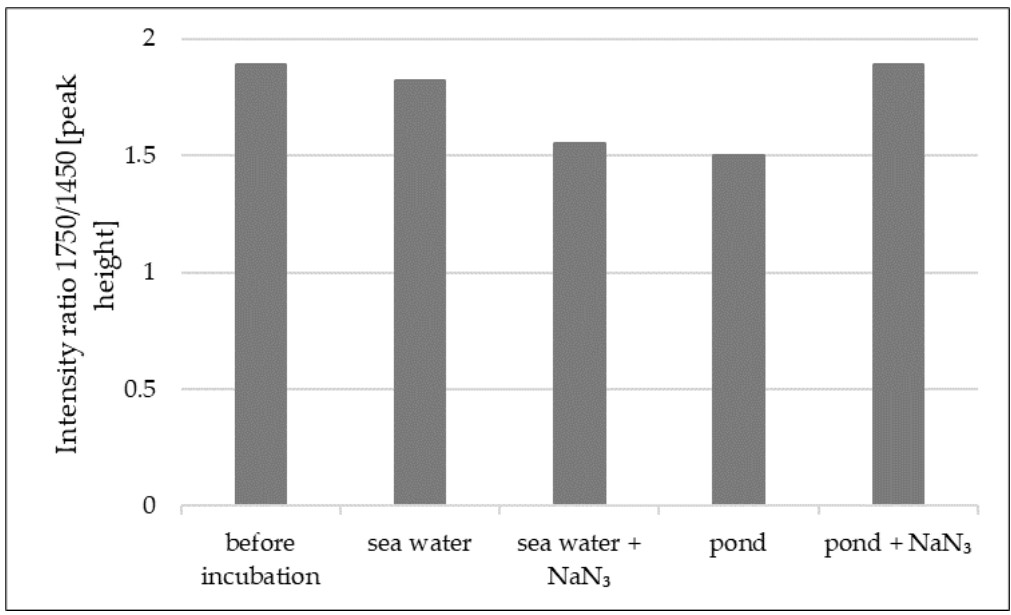

**Figure 5.** The intensity ratio of the band at about 1750 $cm^{-1}$ to band approximately 1450 $cm^{-1}$ for PLA before and after 12 months of incubation in aqueous environments. Source: Own research.

It has been reported that polyesters have three possible linkages: carbon–oxygen ester linkage, carbonyl carbon–carbon linkage, and carbonyl carbon–oxygen, which can undergo scission [46].

Figure 3a shows the spectra of the ATR-FTIR PLA sample before incubation. The main bands detected are indeed those characterizing the PLA structure, for instance: the band centered at 1747 cm$^{-1}$, corresponding to the C=O stretching mode of the ester group and the envelope of bands with a maxima at 1077 cm$^{-1}$ (C–O–C stretching modes), and 1042 cm$^{-1}$ (C–C stretching vibration mode). Another maximum at 1127 cm$^{-1}$ is assigned to the CH$_3$ rocking mode. Weaker bands at 1450, 1380, and 1360 cm$^{-1}$ are due to asymmetric and symmetric C–H deformation modes in the CH$_3$ and CH groups, whose corresponding stretching modes are detected in the high-frequency region of the spectrum at 2994, 2943, 2883, and 2855 cm$^{-1}$ (weak). A few of the other very weak bands around 1265 cm$^{-1}$ should also be assigned to the CH deformation modes. Similar findings have been reported in the literature [46–48].

Two bands related to the crystalline and amorphous phases of PLA were found at 864 cm$^{-1}$ and 745 cm$^{-1}$. According to the literature, the peak at 864 cm$^{-1}$ can be assigned to the amorphous phase and the peak at 745 cm$^{-1}$ to the crystalline phase [47,49]. Other authors analyzed the changes in the molecular and supramolecular structures of PLA by FTIR and indicated that the peaks located at 921 cm$^{-1}$ are associated with ordered regions of $\alpha$ and $\alpha'$ polymorphic forms, while the peaks located at 957 cm$^{-1}$ correspond to amorphous regions of PLA [50].

During hydrolytic degradation, the amorphous regions are degraded first, showing a visible decrease in the peak intensity at 957 cm$^{-1}$. This leads to an increase in the degree of crystallinity with the progress of degradation, which is visible as a peak at 921 cm$^{-1}$. Giełdowska et al. proposed the analysis of the overall crystallinity index by measuring the absorption of the 921 and 957 cm$^{-1}$ bands and determining the overall proportion of the ordered areas in the material [50]. Therefore, the overall crystallinity index values of PLA after incubation in all of the aqueous environments were calculated using the equation proposed by Giełdowska et al. Only a slight increase in the crystallinity index values of PLA (approximately 1−2%) were noticed after 12 months degradation in natural and laboratory environments.

It is known that the degradation of polyesters occurs primarily in the amorphous part of the material; hence, its remaining degraded matter is characterized by a high overall crystallinity index [50].

It is difficult to state that the slightly noticeable change in the degree of crystallinity PLA after incubation in all environments is the result of the loss of the amorphous phase in the hydrolytic degradation process. This result should be confirmed by further analyses.

Figure 3b shows the spectra of the ATR-FTIR PLA sample after incubation in different aqueous environments. It is well-known that ester bonds are relatively easy to hydrolyze and are susceptible to enzymatic (microbial) cleavage. The characteristic bands of groups potentially susceptible to hydrolysis were analyzed. It is clearly visible that the structural changes in PLA after incubation in both natural environments (seawater and the pond) are dissimilar.

The absence of a band indicating the presence of OH groups in the ATR-FTIR spectrum of the PLA samples incubated for 12 months in seawater may be due to the different biotic and abiotic conditions in both natural environments.

The spectra of the PLA samples incubated for 12 months in the pond showed a characteristic broad peak in the region of 3000 cm$^{-1}$–3700 cm$^{-1}$. It indicates the formation of OH groups as a result of ester group degradation. Moreover, the spectra of the PLA were also incubated in seawater, and NaN$_3$ showed a characteristic broad peak in this region (Figures 3b and 4).

In the literature, Cuadri et al. and Moliner et al. analyzed the PLA chain scission reactions and proposed an analysis of the relative intensities of the ester band, which is the intensity of the band at 1745 cm$^{-1}$ (C=O stretching), normalized using the intensity

of the band at 1450 cm$^{-1}$ (CH$_3$ deformation mode) as the internal standard PLA [48,49]. Degradation by hydrolysis should lead to a decrease in this ratio, explained by the random disappearance of the ester linkage in the chain and the corresponding formation of carboxylic groups in oligomers and monomers [48].

Therefore, this parameter was evaluated to investigate possible changes in the chemical structure of PLA after 12 months of incubation in aqueous environments. The intensity ratio [intensity of carbonyl band (about 1750 cm$^{-1}$)/intensity of CH band (about 1450 cm$^{-1}$] was calculated for the PLA samples considering peak height (Figure 5).

Considering the ratio referred to the height of the main ester peaks, degradation in some aqueous environments decreases this ratio, suggesting hydrolysis of the groups, and this is especially evident after 12 months of incubation in the pond and seawater with NaN$_3$.

The hydrolytic degradation of PLA can proceed through two different mechanisms: surface reactions and bulk erosion [18,51]. The former type of degradation generally proceeds much faster than that of the latter type and is probably connected with the presence of microorganisms in natural water [11]. Bulk degradation occurs when water molecules penetrate into the polymer network, causing the hydrolysis of the chain. The hydrolytic degradation proceeds inside the molecular structure. This mechanism was assigned to the internal autocatalytic effect of the carboxyl end groups. In aqueous solutions, the hydrolytic degradation of PLA proceeds via random cleavage of the ester bond [51]. Comparing the results of PLA incubation in freshwater, only a small mass loss of samples was noticed after 12 months of incubation in used degradative solutions (Figure 2). Results of the intensity ratio of the band at approximately 1750 cm$^{-1}$ to the band at approximately 1450 cm$^{-1}$ for PLA before and after incubation in the pond (Figure 5) indicate that chain was cut through surface erosion but created short chains were not moved from polymer network, so the sample mass was unchanged. On the other hand, the results on degradation in seawater alone indicate a very low susceptibility of PLA to degradation in this aqueous environment. The lack of changes in the intensity ratio of the band at approximately 1750 cm$^{-1}$ to the band at approximately 1450 cm$^{-1}$ for PLA incubated in natural seawater suggested that PLA is not degraded via hydrolysis (Figure 5). It also seems that the term of 16 months of incubation in this natural environment was too short for the hydrolytic degradation of PLA proceeds via the random cleavage of the ester bond [18]. In sum, the degradation processes in both natural aqueous environments appear to be very limited. Despite some changes in the chemical structure, especially after treatment in the pond, the PLA seems to be largely resistant to hydrolysis during 12 and 16 months of incubation.

Comparing the results of both laboratory environments, we can state that they differ. The sodium azide was added to the water to eliminate the activity of living organisms and to evaluate the vulnerability of the PLA to chemical hydrolysis in both waters. The results shown in Figure 5 suggest the chemical hydrolysis of ester groups occurs only in seawater with NaN$_3$ because the lower ratio due to the height of the main ester peaks was noticed. It could be explained by the higher surface tension of saltwater than freshwater and, consequently, the difficulty in penetrating the water inside the molecular structure of the polymer and the leaching of the acidic products of PLA hydrolysis. This could cause a greater extent of autocatalysis in seawater with NaN$_3$ [52,53]. The influence of immersion in the pond and seawater on the mechanical properties of the investigated polymer was also estimated. Changes in the mechanical properties of the PLA samples after degradation in aqueous environments were examined by the measurement of their tensile strengths at breakdown and elongation. The results are presented in Figures 6 and 7.

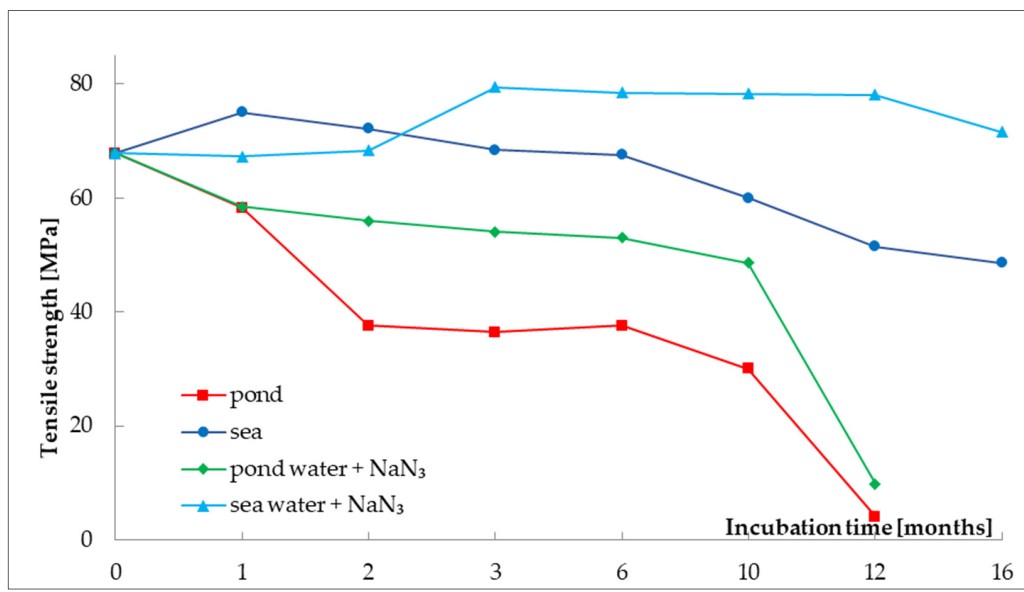

**Figure 6.** Tensile strength [MPa] of PLA samples after incubation in different aqueous environments. Source: Own research.

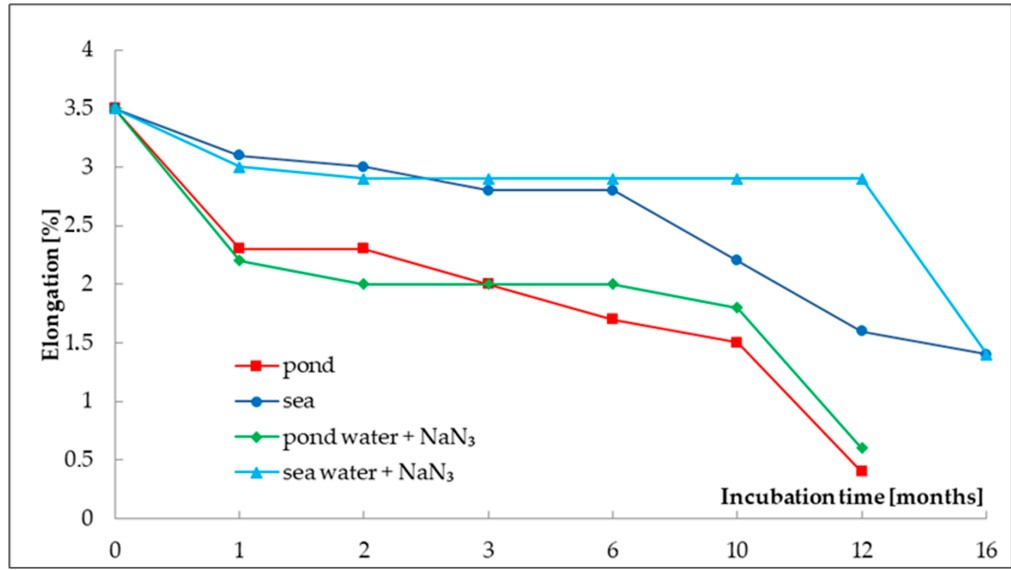

**Figure 7.** Elongation [%] of the PLA samples after incubation in different aqueous environments. Source: Own research.

There is no correlation between weight changes and mechanical properties because the changes in tensile strength after incubation in all aqueous environments are more visible than the changes in weight (Figure 2).

In analyzing the changes in tensile strength, the nature of the aqueous medium significantly affects the polymer properties. We can observe that the tensile strength decreased steadily for the PLA samples during degradation in the pond (to 4.1 MPa), which could be explained by the activity of specific living organisms in freshwater. In contrast to the degradation of PLA in pond water, the increase of tensile strength was observed at the beginning of incubation in the Baltic Sea. It could be caused by an unexpected additional crosslinking as a synergistic action of biotic and abiotic factors in the marine environment. At the end of the exposure of PLA in the Baltic Sea, a slight decrease in tensile strength

(to 48.6 MPa) was noted (Figure 6). A possible reason is the role of mineral salts in this environment, which can influence hydrolytic degradation [14].

The incubation of PLA in all of the aqueous environments caused the band corresponding to the C=O of the ester group to shift slightly to the higher wavenumbers (Figure 3). It indicates that after degradation, some of the hydrogen bonds between the ester groups had dissociated, and several more bands corresponding to the $C=O_{free}$ groups became more visible. The presence in this region of the spectrum of bands corresponding to the vibrations of the free C=O groups was indicated by other authors [54]. The chain of PLA becomes more mobile and creates the possibility of good ordering and, consequently, an increase in crystallinity. Unexpectedly, our study indicates only a slight increase in the overall crystallinity index of PLA (approximately 1–2%). Thus, the dissociated hydrogen bonds between the ester bonds are likely more responsible for the decrease in tensile strength.

The curves in Figure 7 show that the elongation of PLA samples slowly decreased during incubation in all of the aqueous environments. A greater decrease in elongation of the samples was observed after degradation in the pond (from 3.5% to 0.4%) than after degradation in seawater (from 3.5% to 1.4%). The results of the decrease in elongation of the samples confirm the results of the decrease in tensile strength of the polylactide samples.

The observed changes in mechanical properties of the PLA during 12 and 16 months of incubation, respectively, in the pond water and the Baltic Sea, indicate that PLA is also susceptible to fragmentation in these environments. As a result, before its environmental degradation, it will cause the formation of secondary microplastics in natural aquatic environments.

Finally, the degradation of the investigated PLA samples was evaluated on the basis of changes in polymer morphology (Figures 8 and 9). After incubation in the aqueous environments, the analyzed samples were not homogeneously destroyed. There were different images of PLA structure depending on the place of analysis. The most repeated images observed under the metallographic microscope were done.

The morphology of the blank PLA sample (Figure 8a) was homogenous without any oriented phase. The first microscopic observations indicate a similar susceptibility of PLA to biological degradation in both natural aqueous environments (3 months).

If we look at the micrographs (Figure 8b,c), we can see the first slight differences in the surface degradation of the PLA after 6 months of incubation in the pond and seawater. After the following months of exposure, the PLA surface in natural environments is slowly degraded and characterized by the appearance of many dark places of different sizes and shapes (Figure 8d,e). The effects are visible for the PLA incubated both in the pond and sea, but the size of the dark places is much smaller after degradation in the natural pond (Figure 8e,f). The incubation of PLA samples in the Baltic Sea leads to the formation of a distinct black area on the surface, which likely corresponds to the formed agglomeration of microorganisms' biofilm on the surface (Figure 8h) [15].

It is known that microorganisms can form biofilm on the polymers' surface in every environment. The biofilm participates in polymer degradation, but its abundance on the surface is dependent on bacterial species, temperature, pH and incubation time [11,32,55].

Despite the lack of essential weight changes in the PLA samples (Figure 2), the microscopic observations show that the degradation of PLA samples in natural aqueous environments proceeded slowly and is slightly visible.

The degradation process of the PLA samples in both laboratory waters with sodium azide was not very significant (Figure 9a,b). There were no evident changes in the PLA morphology after 12 months of incubation in pond water with $NaN_3$ and after 16 months of incubation in seawater with $NaN_3$. Slight changes were shown only during chemical hydrolysis, which was confirmed by changes in the chemical structure (Figure 5), especially after incubation in seawater with sodium azide.

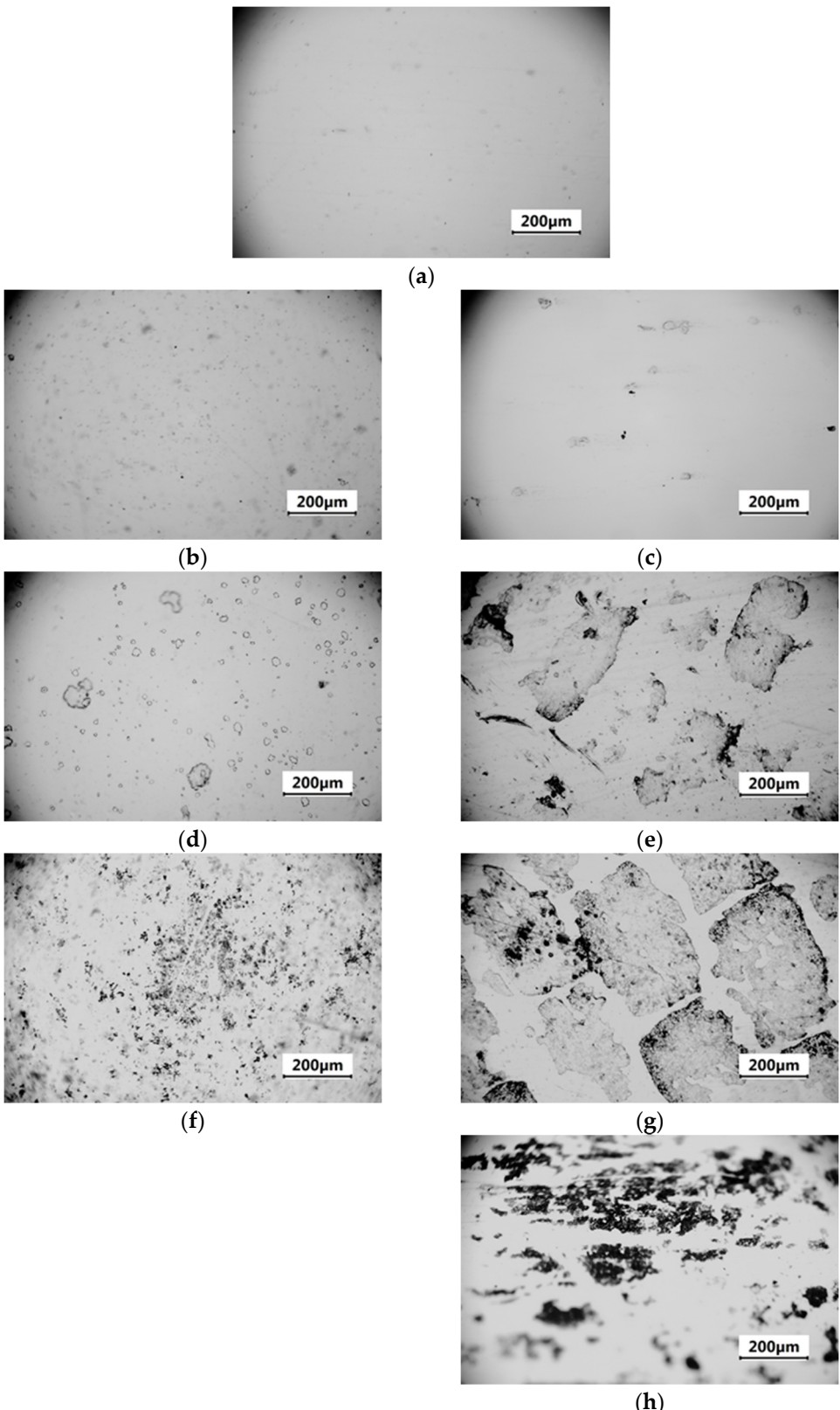

**Figure 8.** Micrographs of PLA samples before and after incubation in natural aqueous environments: (**a**) before degradation; (**b**) 3 months in pond water; (**c**) 3 months in seawater; (**d**) 6 months in pond water; (**e**) 6 months in seawater; (**f**) 12 months in pond water; (**g**) 12 months in seawater; (**h**) 16 months in seawater. Source: Own research.

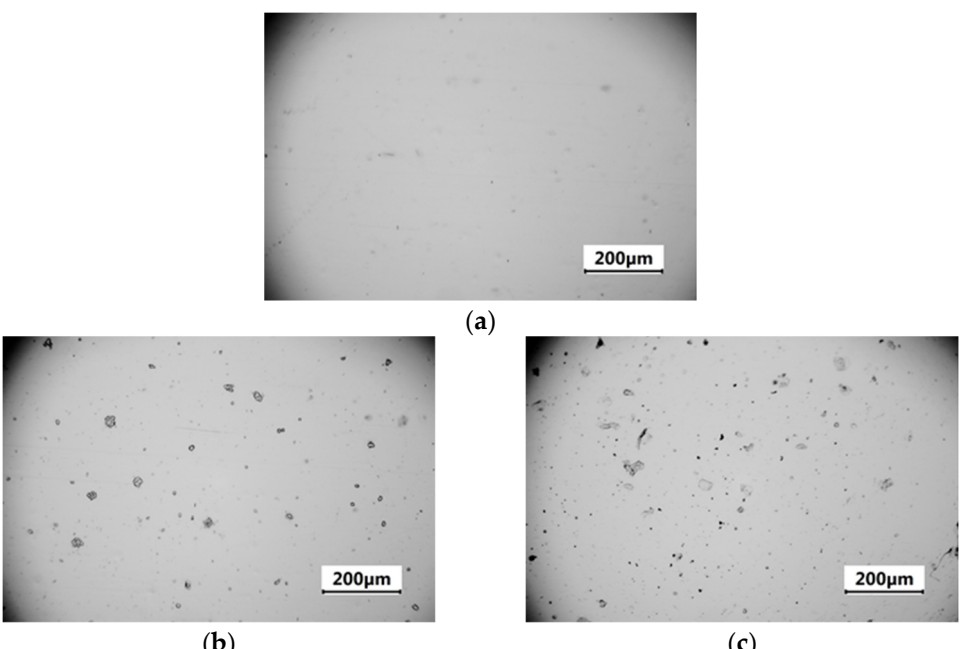

**Figure 9.** Micrographs of PLA samples before and after incubation in aqueous environments with sodium azide: (**a**) before degradation; (**b**) 12 months in pond water; (**c**) 16 months in seawater. Source: Own research.

## 4. Conclusions

Biodegradable polymers in natural environments are often not as "biodegradable" as they are in laboratory tests. Degradability always depends on the specificity of the environment. Temperature, pH, oxygen content, and concentrations of microorganisms are various in natural environments, resulting in different biodegradation rates.

In summarizing the experimental results, the following conclusions can be derived with regard to the degradability of polylactide in aqueous environments:

- The results of macro and microscopic observations and weight changes indicate a slow degradation process in the pond and in the Baltic Sea.
- The lack of significant changes in polylactide indicates that the investigated polylactide requires more than 12 and 16 months to be destroyed in the natural environments. It is due to the low temperature of environments and the kind of living macro- and microorganisms in "real" aqueous environments.
- FTIR spectroscopy analysis evidenced that very limited degradation phenomena occur in both natural environments, and the PLA seems to be resistant to hydrolysis during the monitored period.
- The changes in the mechanical properties of polylactide indicate a gradual degradation in the pond and the Baltic Sea, but the pond results are more visible.
- Insignificant changes in weight and polylactide surface after incubation in laboratory tests may be explained by slow chemical hydrolysis of ester bonds, but the measurements of the molecular weight of the investigated polymers should confirm these suggestions.

More research and deeper knowledge is needed in respect of the degradation of polylactide in different natural aqueous environments. Polylactide is not a polymer that biodegrades rapidly in every kind of ecosystem. Although polylactide is the most prominent representative of biodegradable polymers and degrades in compost rapidly, it is resistant to degradation in aqueous environments (pond and sea).

In this paper: we have shown that the hydrolytic degradation of polylactide in natural aqueous environments is a complex mechanism due to the polymer properties and its

temperature and microbial sensitivity. This is not, in itself, surprising, as polylactide is a biodegradable polymer under specific conditions. To examine the environmental degradation of polylactide in a pond and the Baltic Sea water, a longer term of polylactide exposure is required to estimate the potentially adverse ecological impacts of degradation of polylactide products and the small (microscopic) plastic particles that can arise via its fragmentation.

**Author Contributions:** Conceptualization, K.K. and A.H.; methodology, K.K. and A.H.; investigation, K.K. and A.H.; writing—original draft, K.K. and A.H.; writing—review and editing, K.K. and A.H. All authors have read and agreed to the published version of the manuscript.

**Funding:** The article presents results developed in the scope of the research project "Monitoring and analysis of the impact of selected substances and materials in terms of environmental protection", supported by Gdynia Maritime University (project grant no. WZNJ/2022/PZ/10), as well as supported by the project "Marine port surveillance and observation system using mobile unmanned research units" grant no. NOR/POLNOR/MPSS/0037/2019-00.

**Data Availability Statement:** Data sharing is not applicable.

**Conflicts of Interest:** The authors declare no conflict of interest.

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
