# Peer review of "Degradability of Polylactide in Natural Aqueous Environments"

_water, doi:10.3390/w15010198_

Round 1
Reviewer 1 Report
In this paper author show the interesting results of investigation of behaviour of polylactide during degradation in natural aqueous environments. However, I have following recommendations for the author/s:
Main comments
The experiment is interesting but in the Reviewer opinion could be conducted for the longer time. The “Weight changes” which should be named “Mass loss” change only 5% for the sample degraded in sea water+NaH3. This results show the strong resistivity of PLA for the hydrolytic degradation under temperatures below glass transition. In the higher temperatures the degradation is more rapidly, pleas see DOI: 10.1021/acs.macromol.2c01778.
The more visible changes was observed in the case of mechanical parameters which in the Review opinion as the effect of molecular and supramolecular changes of PLA what could be presented by the FTIR results. FTIR results are presented and interesting aspects have been discussed, but there are still untapped areas like crystallization
However, the experiment is interesting and after major changes (few important points) with a more in-depth analysis of the results can be published in the WATER journal.
Major revisions
1. The presented results in figures 2, 6, and 7 should be more readable and show the real changes of investigated parameters. In the reviewer opinion the point form or points with the polygonal chain should be applied. The presented results as the smooth curves created the main question: Why are majority of experimental points (I hope mean values) joint by smooth curves and how was statistical criterion for these curves creation?
2. FTIR result are interesting but the main missing is the determination or analysis of change in supramolecular structure of PLA. E.g. the crystallinity index could be calculated for the bands 870 cm-1, 920 cm-1 and 950 cm-1, please see DOI: 10.1021/acs.macromol.2c01778. The analysis of crystallinity index will be explain the mechanical changes of studied foils. The higher crystallinity with the α’ to α phase transition of PLA crystalline form (observed as the changes of 2nd derivative of 1750 cm-2 band) affect the decrease the tensile strength of material please see please see DOI: 10.1177/00405175166733.
Minor revisions
1. Tittle of proposed paper is: “Behaviour of Polylactide During Degradation in Natural Aqueous Environments”. In the Review opinion the title should be more scientific. The term “behaviour” could be change to the “properties” or “structural changes” but the short title like e.g. Polylactide Degradation in Natural Aqueous Environments, could be better.
2. Material Section, PLA products of NatureWorks with trade mark Ingeo are a whole range differing in molecular weight and application, please provide the symbol. Probably it was 4 series, maybe Ingeo 4032D but you need to check it and complete it in Material section. It is important information.
Author Response
Response to Reviewer 1 Comments
We would like to thank the Reviewer for helpful comments to improve the manuscript. After a careful consideration of all comments, we have produced a new revised version.
Point 1 Major revisions:
The presented results in figures 2, 6, and 7 should be more readable and show the real changes of investigated parameters. In the reviewer opinion the point form or points with the polygonal chain should be applied. The presented results as the smooth curves created the main question: Why are majority of experimental points (I hope mean values) joint by smooth curves and how was statistical criterion for these curves creation?
Response: According to sugestion of Reviewer, we changed presentation of results in figures 2, 6, and 7. The points with the polygonal chain are applied. We agree that currently they are more readable and show the real changes of investigated parameters.
Point 2 Major revisions:
FTIR result are interesting but the main missing is the determination or analysis of change in supramolecular structure of PLA. E.g. the crystallinity index could be calculated for the bands 870 cm-1, 920 cm-1 and 950 cm-1, please see DOI: 10.1021/acs.macromol.2c01778. The analysis of crystallinity index will be explain the mechanical changes of studied foils. The higher crystallinity with the α’ to α phase transition of PLA crystalline form (observed as the changes of 2nd derivative of 1750 cm-2 band) affect the decrease the tensile strength of material please see please see DOI: 10.1177/00405175166733.
Response: We would like to thank Reviewer for indicating other possibilities of using FTIR analysis, which we will certainly use in our next detail explanation of the FTIR results.
In accordance with Reviewer's guidelines the crystallinity index was calculated and disscussed for the bands indicated in the paper DOI: 10.1021/acs.macromol.2c01778 (lines: 266-286 of the manuscript).
In our paper only before and after 12 months incubation in all aqueous environments the chemical structures and chain conformations of the PLA were characterized with FTIR. In our study we estimate changes of the overall crystalinity index after 12 months of incubation time.
Considering the slight increase of the overall crystallinity index of PLA (aproximetely 1-2%), masured by means of FTIR, difficulty to discuss its infuence on the tensile strength. However, the part discussion of results was corrected (lines: 378-387 of the manuscript).
Point 1 Minor revisions:
Tittle of proposed paper is: “Behaviour of Polylactide During Degradation in Natural Aqueous Environments”. In the Review opinion the title should be more scientific. The term “behaviour” could be change to the “properties” or “structural changes” but the short title like e.g. Polylactide Degradation in Natural Aqueous Environments, could be better.
Response: We changed the tittle of manuscript due to Reviewer comment. We chnged to “Degradability of Polylactide in Natural Aqueous Environments”. According to sugestion the term “behaviour” was changed in lines: 90, 92, 443.
Point 2 Minor revisions:
Material Section, PLA products of NatureWorks with trade mark Ingeo are a whole range differing in molecular weight and application, please provide the symbol. Probably it was 4 series, maybe Ingeo 4032D but you need to check it and complete it in Material section. It is important information.
Response: We agree with Reviewer that PLA products of NatureWorks with trade mark Ingeo are a whole range differing in molecular weight and application. The subject of our investigation was a commercial multilayer film BIO PLA™. BIO PLA™ is one product from the BioPak. Raw material of this film was manufactured from NatureWorks® PLA, but we do not the exact type of PLA. We know that it is important information, but the manufacturer has not made it available to us. We only received an information brochure, which we are sending as an attachment.

Reviewer 2 Report
The authors conducted a full-workload study. Using these research methods in the manuscript, the authors can draw reasonable conclusions. Unfortunately, however, this manuscript has major issues that cannot be ignored. First, the selection of subjects in this study is not universal, and the parameter selection is difficult to provide a reference with wide use value. Secondly, there are also problems with the writing method of this manuscript, and neither abstracts nor conclusions have precise numbers. Finally, as a typical degradable plastic, PLA's degradation process is perhaps not the most noteworthy. I believe that the level and conditions of the authors are sufficient to complete the high level of work, but I regret that this manuscript is not suitable for publication in current form.
Author Response
Response to Reviewer 2 Comments
Overall comments:
The authors conducted a full-workload study. Using these research methods in the manuscript, the authors can draw reasonable conclusions. Unfortunately, however, this manuscript has major issues that cannot be ignored. First, the selection of subjects in this study is not universal, and the parameter selection is difficult to provide a reference with wide use value. Secondly, there are also problems with the writing method of this manuscript, and neither abstracts nor conclusions have precise numbers. Finally, as a typical degradable plastic, PLA's degradation process is perhaps not the most noteworthy. I believe that the level and conditions of the authors are sufficient to complete the high level of work, but I regret that this manuscript is not suitable for publication in current form.
Response :
We would like to thank the Reviewer 2 for comments to improve the manuscript. After a careful consideration of all comments, we would like to provide responses to the owerall your comments.
We selected two possible natural environments in our country (The Baltic Sea and a pond), where in the future can be a place for accumutation of polymers waste. The natural environments of both seawater and freshwater are unpredictable and degradation process depends on many biotic (microorganisms, enzymes ect.,) and abiotic (pH value, temperature ect.) factors. All of which cannot be well controlled and demand careful monitoring. Incubation the polymer in a pond, or the sea, provides a realistic environment where plastic litter could end up. The Baltic Sea is specific environment, it is small, closed and cold, so it can not be considered as universal environment. The degradation of biodegradable polymers in freshwater is usually analyzed in rivers and lakes, so it was interest to choose the pond. Unfortunately, a lot of used plastics end up in surface water and pollute the natural aqueous environment. Therefore, we think that the present study is important and of general interest (especially in the north part of Europe).
The concept of this paper was based on study the behaviour of PLA in “real” environments (under natural weather depending conditions) to assess whether it can caused creation of secondary microplastics in these environments. The main goal of this paper was the estimation of the susceptibility to degradation of PLA during incubation in two natural environments (the pond and The Baltic Sea). Considering above the main aim of the work, the precise numbers were not introduced neither in abstract and finally conclusion. All precise numbers from our study are detailedly presented and described in the section Results and Discussion. Both abstract and connclusions were prepared according to Instructions for Authors.
We do not agree that PLA's degradation process is not the most noteworthy. The PLA is considered as a biodegradable polymer produced on the bases of renewable raw materials, its use is attractive and may help to reduce ecological problems related to waste plastics. This results show the strong resistivity of PLA for the hydrolytic degradation under temperatures below glass transition. In the higher temperatures the degradation is more rapidly, but not in natural water environments where the temperature is lower. The our results indicate clearly that PLA is nearly not degraded in the studied natural waters under natural weather depending conditions. The obtained results indicate that although this the most known biodegradable polymer degrades in compost rapidly but in selected natural aqueous environments is resistant to degradation.These negative results confirm the known limitations of PLA degradability and they are very important for a correct waste management of biodegradable plastics. For example from the practical point of view the label “biodegradable” for PLA should have a clear sign of the environment.
Despite the lack of your precision remarks indications we have produced a new revised version according to comments of the all Reviewers .

Reviewer 3 Report
see report added

Author Response
Response to Reviewer 3 Comments
We would like to thank the Reviewer for helpful comments to improve the manuscript. After a careful consideration of all comments, we have produced a new revised version.
Point 1 (bullet point 1):
Table 1 indicates the oxygen concentrations vol(O2)/vol(liq), [cm3/dm3]. As the gas volume depends strongly on pressure and temperature, I propose to communicate the O2-concentration in mass/vol: [mg/ dm3].
Response: We agree with the Reviewer that the concentration of oxygen as a volume of gas depends strongly on pressure and temperature. Also, the oxygenation of the surface layer of the sea undergoes normal seasonal changes due to changes in water temperature and the solubility of oxygen in it, which are superimposed on the effects of the photosynthetic process. Oxygen concentration expressed in the unit mg/dm3 is often used, as well as cm3/dm3. In our work, this part of the study was carried out by The State Environmental Monitoring, Inspectorate of Environmental Protection in Gdynia. We received all the Baltic Sea parameters from this Institute, which we indicated in Table 1.
Point 2 (bullet point 2):
Figure 2: Is there any plausible explication for the accelerated mass loss in sea water between the 12th and 16th month?.
Response: Based on study of literature and our previous studies conducted on the degradation of various biodegradable polymers in natural environments, temperature is one of the possible factors affecting the rate of biodegradation.
The observed slight acceleration of PLA mass loss in seawater between 12 and 16 months is most likely due to degradation carried out during the summer months (May-November), where seawater temperatures were higher than in the winter months. Regarding reviewer's comment, this explanation was also included in the manuscript (line: 223-226).
Point 3 (bullet point 3):
The experimental procedure of addition of sodium azide to sea water and pond water must be explained in detail. As the polymer samples are placed in perforated baskets dipped in open water, NaN3 will be washed out rapidly.
Response: The sodium azide was added to water taken from the natural environment of pond and sea and placed in a 60-liter glass aquarium in the laboratory. In the laboratory, both waters were oxygenated and monitored for temperature and pH. The sodium azide was added to eliminate activity of living organisms and to evaluate PLA vulnerability to the chemical hydrolysis.
Only in the natural environment of The Baltic Sea and pond, the samples were placed in perforated baskets to allow free flow of water, microorganisms and enzymes into the degraded material (here, sodium azide was not added).
We explained the experimental procedure in detail in section 2.1 Environments of the manuscript.
Point 4 (bullet point 4):
The results shown in Fig. 5 concerning the influence of NaN3 are quite contradictory. Please discuss the inhibitory influence in pond water and the accelerating effect of NaN3 in sea water.
Response: We agree with Reviewer that the results presented in Fig. 5 are different in both laboratory environments and can be seen as contradictory. In our paper the sodium azide was added only to eliminate the activity of living organisms and to evaluate the vulnerability of the PLA to chemical hydrolysis respectively in salt and fresh water. This slight increase in hydrolysis in salt water (the ratio referred to the height of the main ester peaks decrease from 1.89 to 1.55) compared to fresh water was explained by autocatalytic bulk erosion of PLA and disscussed in the manuscript (line: 347-356).

Round 2
Reviewer 2 Report
Despite my concerns was not addressed by the revisions, the authors struggled to revise their manuscript and the quality of the manuscript improved. Therefore, I do not oppose the publication of this manuscript.
Reviewer 3 Report
no remarks